# Why Are Child and Youth Welfare Support Services Initiated? A First-Time Analysis of Administrative Data on Child and Youth Welfare Services in Austria

**DOI:** 10.3390/children10081376

**Published:** 2023-08-11

**Authors:** Katja Haider, Stefan Kaltschik, Manuela Amon, Christoph Pieh

**Affiliations:** Department of Psychosomatic Medicine and Psychotherapy, University for Continuing Education Krems, 3500 Krems, Austriamanuela.amon@donau-uni.ac.at (M.A.); christoph.pieh@donau-uni.ac.at (C.P.)

**Keywords:** child welfare, child endangerment, foster care, children, adolescents, administrative data, Austria

## Abstract

Even if numerous children and young people are looked after by child and youth welfare, there are only a few scientific studies on the reasons for this support. The aim of this retrospective descriptive study was to examine the reasons why child and youth welfare was initiated. Therefore, administrative data, collected by the Lower Austrian Child and Youth Welfare Service, from the year 2021 will be presented. On the one hand, the frequencies of the different justifications provided by the social workers and, on the other hand, whether these are primarily based on problems of the parents/caregivers or the children are reported. In 2021, a total of 7760 clarifications of child welfare endangerments were initiated. The descriptive statistical analyses showed that the most frequent concerns were parental overload (49%), behavioral issues (10%), and difficult economic conditions (9%). Although a classification according to the caregiver or child level cannot always be clearly distinguished, there is a trend that in many cases (84% to 99% depending on the type of support) the problems lie at the caregiver level. Further studies are necessary so that the care of such vulnerable groups of people will be better supported by scientific findings.

## 1. Introduction

Numerous studies have shown that child endangerment is a strong risk factor for the psychological and social development of a child. Engler et al. [1] reported that mental illnesses (ADHD, depression, and anxiety disorders) and behavioral problems were significantly overrepresented among those affected compared to children who were not in full-time care. Furthermore, these children and adolescents showed a higher rate of suicide, suicide attempts, and suicidal thoughts [1]. It was also noted that the mental health of these children was influenced by the type of child endangerment with child neglect, physical abuse, and sexual abuse being reported as the strongest predictors. Negative consequences were also observed in the long term. Adults who were exposed to sexual, physical, or psychological abuse during childhood had significantly poorer mental health (e.g., eating disorders, alcohol abuse, suicidality, depression, and lower self-esteem), lower educational attainment, and lower socioeconomic status. Divorces, pregnancies before the 19th birthday, and an increased probability of child and youth welfare for one’s children were also more common. These effects were all the more fatal when there was an accumulation of several forms of risk [2,3]. The potential indirect costs of mental illnesses (e.g., reduced quality of life, secondary disorders, comorbidities, burden for family members, increased need for social support, future inability to work, unemployment or early retirement, or increased risk of crime) should not be ignored when discussing child endangerments as contributing to mental illnesses.

Recent research also showed that child vulnerability rarely occurred independently and in isolation. The classification of child endangerment in only one category therefore insufficiently reflects the reality and is therefore not very meaningful for research purposes. It appears that the severity of child maltreatment is a more reliable indicator of the course of events than the simple categorization [4,5].

In Austria, the child and youth welfare law [6] provides several measures to safeguard or restore the well-being of children and adolescents. In Lower Austria, these measures are implemented by the Lower Austrian Child and Youth Welfare Service (LA CWS). The CWS provides a wide range of measures, which are adapted to the severity of child endangerment, ranging from low-threshold advice to taking over parental authority to ensure the child’s well-being and safety. All of the applied measures are documented by the responsible social worker in a case documentation system for primarily administrative purposes. Official reports of support services provided by the CWS in Austria are published by Statistics Austria [7]. Related statistical data and developments in Lower Austria are published in an annual report [8].

In recent years, the benefits of using administrative data for research purposes has also been discussed in the field of child and youth welfare [9,10]. Although administrative data are not primarily collected for research purposes, they can often be used profitably in a scientific context. Since administrative data are usually collected according to criteria other than data collected explicitly for research purposes, hurdles and peculiarities in data access, data processing, and data analysis often occur. The process of accessing, processing, and analyzing data, with all its difficulties, is described in detail in a study evaluating administrative data in the field of child and youth welfare in several states in the USA [11]. The authors discuss problems and solution approaches in the areas of data access, data availability, and data comparison, as well as the operationalization and coding of research-relevant variables, and conclude that the knowledge gained from the analysis of administrative data usually outweighs the effort [11]. The importance of using administrative data for research in the field of child and youth welfare has also been recognized despite its challenges. Nonetheless, the potential benefits of using administrative data for research in this area outweigh the difficulties, and a variety of data linkage methods have been identified that can facilitate this process [12].

Although administrative data provide a rich source of information about the characteristics of children and families who come into contact with the child welfare system, the support services they receive, and the outcomes they experience, so far, few scientific studies approach this topic. In general, little is known from a scientific perspective about why child and youth services are initiated. While some studies focused on the types of referral to child and youth welfare services (e.g., police, schools [13], dental health personnel [14], or social service personnel [15]), certain subgroups (e.g., mothers with mental health issues [13] and parents with intellectual disabilities [15]), or individual allocations (e.g., neglect, physical abuse, and sexual abuse [16], substance misuse [17,18], or poverty [19,20]), there is a lack of consideration of the full range of reasons for child and youth welfare involvement, especially on the level of the children. Therefore, the aim of the present study was to exploratively examine the reasons why child and youth welfare services were initiated, whether more justifications were at the caregiver or the child level, and how the justifications and services were distributed between the genders and age groups. Thus, this is also the first scientific investigation of child and youth welfare initiation reasons based on data from Lower Austria. Thereby, the paper makes a significant contribution to initial research on child and youth welfare (especially in Austria), adds to the sparse base of literature on the topic, and demonstrates the usefulness of administrative data for research purposes. The results not only intend to guide policymakers and practitioners to further develop and improve the interventions but also to provide the basis for more in-depth future research in the area of child and youth welfare service initiations.

## 2. Materials and Methods

This study involved an exploratory descriptive analysis of administrative data on the services provided by the Child and Youth Welfare Service in Lower Austria in 2021 and their justifications. The data for this study were collected from administrative sources of the Child and Youth Welfare Service of Lower Austria. The administrative sources included the case documentation system of the Child and Youth Welfare Service (SZF) as well as the service’s annual report. The limitation to a purely exploratory descriptive analysis was based on the structure of the data. Since no case-related data could be accessed for data protection reasons and only the number of services and service justifications in the respective gender and age groups were available, no inference statistical evaluation was possible for the time being.

To obtain summary statistics for the entire services provided by the Child and Youth Welfare Service in 2021, we used the data provided by the service in their annual report. However, this dataset did not include information on the justifications for the specific services, the age groups, and the genders of the children. Therefore, we manually retrieved the data on the ongoing services in combination with their justifications from the case documentation system of the Child and Youth Welfare Service (SZF) between 16 and 22 November 2022.

The SZF system is the primary data repository for child and youth welfare to document and store case-related information for children and families involved in the child welfare system. The system collects a variety of data for each case, including demographic information, use of the services, and justifications for providing the services. However, access to the full database was not granted for our study. Therefore, we used a subset of the database that contained only aggregated data. This dataset provided information on the number of cases in all districts of Lower Austria for specific services, broken down by gender, age group, and justification for the service. Although it was not possible to analyze individual cases, the aggregated data allowed us to examine the frequency of services provided and the characteristics of the children who received them. Since services without a justification were not available from the SZF outputs, the annual report was included to further provide summary statistics for all services administered by the Child and Youth Welfare Service in Lower Austria in 2021. These statistics were used to complement the data collected from the SZF system and to provide a more comprehensive understanding of the services provided by the Child and Youth Welfare Service.

### 2.1. Study Population

We included all ongoing cases that received a support service in 2021. For the analysis of the justifications for services, we used data for the following services: parental support, care home, foster family, and kinship care.

### 2.2. Data Collection Procedure

The analysis included four support services to avert threats to the child’s well-being, which are described in more detail below, including parental support, care homes, foster families, and kinship care:Parental support: This form of support is indicated if there is a risk to the well-being of the child, but it can be prevented by the use of the chosen support measure while remaining in the family or the child’s other previous environment. It should primarily serve to improve the conditions for ensuring the child’s well-being in the family or their previous environment. The category of parental support includes various support services provided by people from outside the family and from different professional groups (e.g., social workers, psychologists, pedagogues, psychotherapists, and pediatric nurses), such as family intensive care, youth intensive care, or afternoon care, which are based on the families’ and children’s individual needs.Care home: Care homes are the intervention of choice when, based on the risk assessment, there is a risk to the well-being of the child or adolescent concerned that can only be averted by caring for the affected child or adolescent outside the family or other previous environment. In Lower Austria, these children are living in communities of groups, where a team of professional pedagogues takes care of them.Foster family: Accommodation in a foster family is an alternative to care homes. It is granted if, due to the age of the child and the problem situation, a suitable foster family can be found.Kinship care: In this form of care, the child or adolescent is raised by a grandparent or other close family member with whom the child or adolescent has a close relationship.

To ensure the accuracy and completeness of the data, cases where no information on the service justification was provided were excluded. This was performed to ensure that only cases with documented reasons for service provision were included in the analysis. The exclusion criteria were applied during the data retrieval process and were based on information available in the SZF system. The exclusion of cases that did not meet the inclusion criteria helped to ensure the accuracy and relevance of the data used for the analysis. To present a complete picture of the ongoing support services via the data analysis, data from the year 2021 were used. Since the data retrieval took place in the fall of 2022, the case documentation for that year would not yet have been complete.

### 2.3. Data Analysis Procedures

We carried out 481 separate queries from the SZF individually, based on the district and type of support service justification, to obtain all relevant service combinations. Due to access restrictions, the process could not be automated, and the 481 files were saved locally in .xlsx format. Each justification*district file then contained frequencies on the respective services in the age and gender groups. To merge the 481 individual files into one, a Python script in Visual Studio Code was created. The data were further analyzed with SPSS (version 29.0.0.0). It is important to note that the data are anonymized and aggregated to protect the privacy of the children and adolescents involved.

When administering a support service in the SZF, a justification is provided by the responsible social worker in one of the below-stated categories. As can be seen in Table 1, we classified these justifications in whether the justification depends on the parents/caregiver or the child. These classifications are based on the semantic concept of the justifications. No statistical procedures were used. Out of 21 possible justifications, 13 were on the caregiver level, and 8 were on the child level. The problems at the caregiver level are inherent to or emanate from the parents/caregivers and have a negative impact on the child, while the problems at the child level are inherent to or emanate from the children, who negatively impact themselves.

Custom crosstabs in SPSS were used for the analyses of the services and service justifications with respect to age groups and gender.

## 3. Results

We used the data obtained from the SZF to analyze the frequency and distribution of the justifications for the ongoing support services of child and youth welfare in 2021 in Lower Austria. The support services themselves and the justifications for utilizing the support services were investigated regarding how these were distributed in relation to various characteristics such as age, gender, and type of support service.

### 3.1. Support Services

The data from the annual report show that a total of 7760 clarifications of child welfare endangerment, which necessarily precede every support service, were initiated by the twenty-four district administrative authorities and clarified by the responsible social work specialists (3958 child welfare clarifications and 3802 other reports) in 2021. If the suspicion of child endangerment was confirmed, a suitable form of support was initiated. The form(s) of support applied is/are chosen by the responsible social workers. A total of around 14,000 services (ongoing and terminated) were provided for children, adolescents, and their families in difficult life situations in 2021, 79% of which were outpatient services where the child or adolescent stayed with their family or in their usual living environment. If outpatient services were considered to not be sufficient to avert the identified endangerment of the child’s well-being, help outside the family was required (a total of 21%), which could be provided within the framework of inpatient help, e.g., care homes, foster families, or kinship care in the long term and short term as part of a crisis accommodation. However, it must be taken into account that the values given relate to the service provided as part of the Child and Youth Welfare Service. For example, a child who received multiple forms of support throughout 2021 was counted for each form of support it received [8].

While, in 2021, 7760 clarifications of child welfare endangerments took place, 4918 ongoing services were provided. The clarifications necessarily precede every support provision to clarify if a minor is at risk, and support is therefore necessary. Out of the 4918 ongoing support services, which were further analyzed in this study, the majority were parental support services (74.3%, *n* = 3653), whereas care homes accounted for 16.6% (*n* = 814), foster families for 5.4% (*n* = 265), and kinship care only for 3.8% (*n* = 186) [8].

To complement the above-described overall child and youth services from 2021, the ongoing service data in combination with the justifications, extracted from the SZF, were analyzed below. When considering the levels of justifications in the analysis, we saw that the majority of all four types of ongoing support services were justified on the caregiver level. As can be seen in Figure 1, hardly any utilization of foster families (1.13%) or kinship care (0.54%) was justified on the child level, whereas about a sixth of care home cases (16.22%) and parental support services (14.21%) were justified on the child level.

With regard to gender, we observed that, overall, slightly more ongoing services were administered for boys (55%, *n* = 2693) than for girls (45%, *n* = 2225) in 2021. Each type of support service was applied more often to boys (parental support: 56%; care home: 53%; foster family: 53%) than to girls except for kinship care. More ongoing kinship care cases were reported for girls (52%) than for boys (48%) in 2021.

In terms of age groups, the data analysis showed that, overall, most ongoing support services in 2021 addressed the age group of six to thirteen (51%, *n* = 2486). Furthermore, 28% (*n* = 1365) addressed the age group of fourteen to eighteen, and 14% (*n* = 669) addressed babies, toddlers, and children up to five years. For some services, the age of the recipients was not indicated (8%, *n* = 398). As seen in Figure 2, in all four support service types, the proportion of 6-to-13-year-olds was the largest, followed by the age group of 14-to-18-year olds, and lastly the group of babies, toddlers, and younger children.

### 3.2. Support Service Justifications

As depicted in Figure 3, which shows the proportions of unclassified justifications across all justifications for ongoing support services in 2021, it can be seen that parental overload was by far the most common justification for the support provided by the Child and Youth Welfare Service. Almost 50% of all services were based on this justification. For all other justifications, the frequency was less than 10%. Except for behavioral issues (9.90%) in second place and school problems (1.89%) in eighth place frequency-wise, all other justifications on the child level gathered at the lower end. The abbreviations C and CG in Figure 3 stand for child and caregiver(s).

As seen in Table 2, by considering gender in the justifications, only slight differences between boys and girls were found. Parental overload was the main reason for receiving support services in both genders. Behavioral issues, the criminal behavior of the child, the violent behavior of the child, and the substance abuse of the child were more likely in male children and adolescents, while we found slightly more justifications due to sexual abuse, underage pregnancy, or the alcohol abuse of the child in female children and adolescents.

Furthermore, differences in the proportions of the two justification levels between the age groups occurred. As seen in Figure 4, the proportion of caregiver-level justifications predominated in all three age groups, but with the increasing age of the children, the proportion of child-level justifications grew. While in the age group up to 5 years only 6% of the ongoing support services were justified on the child level, this number grew to 12% in the age group of 6-to-13-year-olds and to 19% in the age group of 14-to-18-year-olds.

As can be seen in Table 3, parental overload was the most common support service justification in all three age groups. Across all justifications in each age group, the proportion of justifications based on behavioral issues grew with increasing age from 3.74% in 0-to-5-year-olds to 14.21% in 14-to-18-year-olds. Hence, behavioral issues were the second most common justification for child welfare services in 2021. Difficult economic circumstances were in second place for young (0–5 years) and middle-aged (6–13 years) children but occurred less frequently among older (14–18 years) adolescents as justifications for support services. While the proportion of justifications due to difficult economic conditions declined with increasing age, justifications due to caregivers’ divorce or separation increased (0–5 years: 4.63%; 6–13 years: 7.72%; 14–18 years: 8.21%).

## 4. Discussion

In this work, administrative data of the Lower Austrian Child and Youth Welfare Service on the support services and their justifications were evaluated in a systematic and reproducible way for the first time. Most support services provided by child and youth welfare were found to take place due to the caregivers, and children were rarely the problem. The most common justifications for the interventions to avert threats to the children and adolescents were parental overload, behavioral issues, and difficult economic conditions. In addition to gaining insights into service justifications, i.e., why children and adolescents came in contact with the Child and Youth Welfare Service, methods were developed to retrieve and evaluate large amounts of data from the Lower Austrian Child and Youth Welfare Service system. In order to make future evaluations of administrative data more time efficient, it is recommended to automate the step of data retrieval. Since the available data were not collected specifically for research purposes in the first place, they were not in a state in which they could be analyzed with statistical programs commonly used in research immediately. These difficulties in the analysis of administrative data should be considered in future revisions of data collection systems to improve the implementation of automated analysis tools and data quality.

The central aspect of child and youth welfare is the well-being of the child, which is focused on in all support services provided to the children/adolescents and their families. The service provision is therefore tailored to the needs of the individual and available in a timely manner. According to Austrian law, the well-being of children and adolescents is to be secured or restored by providing care that is as mild as possible [6]. Therefore, accommodation in care homes, kinship care, or foster families is only implemented when outpatient measures are not sufficient to avert a child’s endangerment. The fact that the numbers of inpatient services (21%) are relatively low compared to outpatient services (79%) is therefore in line with the principle of the least invasive measure. Outpatient services also have the advantage of better family involvement, and negative effects of out-of-home care (such as relationship disruptions) can be avoided, which in most cases is to the benefit of the children and families and thus promotes the central aspect of child well-being.

The results of this study also showed that the vast majority of all four welfare support services were justified on the caregiver level, which means that either parental support or a form of out-of-home accommodation of the children was necessary due to the parents or caregivers. The individual justification of parental overload was particularly prominent, with nearly 50% across all given justifications. Thus, overload on the part of caregivers was a major contributor to the preponderance of the caregiver level. However, the sole classification of parental overload to the caregiver level must also be viewed critically. On the one hand, caregivers can indeed be the source of parental overload. On the caregivers’ side, there is a connection between parental overload and not only a lack of education, poverty, isolation, and a lack of social support from the immediate environment but also personal insecurity in parenting behavior. On the other hand, one must also acknowledge that parental overload can have its cause in the children themselves, when they, for example, have attachment disorders or show antisocial or delinquent behavior, which can mean an additional challenge for the caregivers. Addiction and other mental illnesses in children can also more quickly lead to overload for caregivers [21]. Therefore, although the classification of parental overload to the caregiver level is the more plausible one since, after all, the overload occurs on the caregivers’ side, it is not a clear-cut one due to the diverse aforementioned reasons for parental overload. Still, it should be noted that, even if the justifications of parental overload are split evenly between the caregiver and child levels, there is still an overall preponderance of justifications at the caregiver level.

Furthermore, the data show differences in the gender ratio within some support service justifications, although due to the data structure, no statements can be made about the statistical significance of the gender and age group differences. A higher representation of boys is, for example, seen in delinquency. This preponderance of male criminality is also reflected in the 2020 conviction statistics of the Austrian judicial crime statistics. Of all convictions, 9% concern male young adults (18–20 years), and 6% concern male adolescents (14–17 years), while only 1% each concern female young adults and female adolescents [22]. Boys and girls differ not only in the frequency of delinquencies but also in their types. Studies show that female adolescents are more likely to be involved in non-aggressive delinquency, such as property crimes [23,24,25] or marijuana use [26,27], while violent and aggressive crimes are less often observed in girls [28]. With an anticipatory outlook on the future risk of delinquency, different forms of abuse, to which children and adolescents in child and youth welfare are frequently exposed, should not be disregarded as it is shown that experiencing marital violence in childhood is predictive of referral to juvenile court. Gender differences also emerge in terms of girls being more likely to be arrested for violent offenses following physical child abuse even though the family risk factor for delinquency is similar for girls and boys [29]. Girls and boys also deal with the experience of abuse differently. Whereas boys tend to externalize with serious and violent delinquency, girls tend to internalize, for example, via suicidal behavior [30,31]. This is also a possible explanation for the gender difference in the delinquency justifications for ongoing support services. Differences in externalizing and internalizing between girls and boys can be explained by societal factors, such as, for example, better relationships with parents and peers and more interpersonal vulnerability in girls and lower susceptibility to self-criticism in boys [32].

The present analysis also showed that the justification “substance abuse of the child” was used more frequently for boys, which is in line with the 2019 data on drug consumption from the European School Survey Project on Alcohol and Other Drugs (ESPAD). The ESPAD survey pointed out that in Austria 25% of male and 19% of female adolescents consumed cannabis in 2019. The annual prevalence of ecstasy, amphetamines, and cocaine was also higher among male than female adolescents [33,34]. Among adolescents, gender differences in drug use can be explained by girls being more highly monitored by their parents and boys being more frequently exposed to deviant peers [35] as well as girls perceiving a greater risk with the use of drugs [36]. The justification “violent behavior” was also used more often for ongoing services provided to boys and is in line with Austria-wide figures, which show that about 90% of those convicted for violent crimes in Austria are men [37].

In contrast, sexual abuse as a support service justification was more prevalent among girls. This pattern was also reflected in the results of the Austrian prevalence study on violence against women and men, which showed that 3.1% of 16–20-year-old young women and 0% of young men of the same age had experienced sexual harassment that subsequently led to unwanted sexual acts. Overall, women were also more likely to have had the following experiences against their will: being touched or fondled intimately (women: 25.7%; men: 8.0%), coercion into sexual acts (women: 13.5%; men: 3.5%), attempted sexual intercourse or penetration (women: 8.9%; men: 2.0%), or actual penetration (women: 7.0%; men: 1.3%). The proportion of women who had experienced this several times was also greater than the proportion of men in all cases [38]. A systematic review also shows that the prevalence of child sexual abuse in Europe is significantly higher among girls with a median of 14.3% (7.8–28.0%) than among boys with a median of 6.2% (14.8–15.2%) [39]. While boys are more likely to experience extrafamilial sexual abuse, girls are more likely to be sexually abused by family members such as their father, stepfather, or other relatives [40]. Easier access to the child within the home and family may be a reason for the gender differences in sexual abuse justifications.

In accordance with the results of the systematic review on European data on emotional and physical abuse, the analysis of the administrative justifications also revealed a slight overrepresentation of girls in emotional abuse justifications and boys in physical abuse justifications. The nearly equal distribution of the two sexes between child neglect justifications is also reflected in the European data of the systematic review [39].

A meta-analysis [41] showed that the prevalence of emotional abuse strongly depends on the type of report (self-report versus third-party report). The estimated prevalence based on self-reports (363 per 1000 boy residents) was more than 100 times higher than that of third-party reports (3 per 1000 boy residents). To determine whether this ratio also applies to Lower Austria, population-representative data with self-report questionnaires should be collected in future studies.

With regard to age, we found an increasing representation with the growing age of the children in justifications referring to the parents’ or caregivers’ divorce or separation. While some studies show that young children could decrease the risk of divorce [42,43,44,45], some even point out that children over the age of 13 could negatively influence the stability of a marriage. A crucial role in this might be the intensive care needs of young children and the possibility of sharing the workload by staying together [46]. Moreover, there is the assumption that the emotional maturation of the children makes the damage of divorce less severe [42].

The fact that behavioral issue justifications are more strongly represented at an older age may be related to the fact that behavioral problems are becoming more frequently relevant for child and youth welfare due to the general conditions at school, which make the behavior issues more easily visible [47,48] not only because requirements such as being quiet, sitting still, or being attentive are required but also because detachment processes and emotional separation in adolescence can lead to conflicts with the parents or caregivers [49,50,51].

Furthermore, 0–5-year-olds were most represented in the justification “difficult economic conditions”, and the representation of the age groups decreased with increasing age. A connection can be assumed between the difficult financial situation of parents with children, especially in babies and toddlers, and parental leave, as well as the increased need for childcare and the accompanying limited possibility of paid employment. On the one hand, parents do not have access to their full salary during parental leave, and on the other hand, there is a general loss of income in the first few years after returning to work, which in Austria amounts to around 14% to 16% in the first year, 6% to 8% in the second year, and 2% to 3% in the fifth year after the return [52]. The lower percentage of older children and adolescents could also be explained by the fact that they can contribute to the household income with their employment and thus buffer difficult economic circumstances. It is shown that especially adolescents from low-household-income families, once employed, contribute relatively more to the family income compared to their peers [53].

The comparison of the results of the administrative data with the data from the above-mentioned statistics and reports not only contributes to a more comprehensive understanding of the social phenomena and problem situations but, in combination, also allows for a better identification of causes and relationships. The matching results also mutually validate each other, and the phenomena are furthermore viewed in a multi-perspective manner. The different data sources also allow for a more informed demand and intervention planning, for example, with regard to sexual violence or drug prevention, and the focus can be placed specifically on risk groups.

### Limitations

For the purposes of this study, it was possible to use data on ongoing services, with a cut-off date of 31 December 2021, and the justifications for these services. Nonetheless, it should be noted here that a bias in the sense of an overestimation in the area of ongoing services cannot be ruled out since several justifications could be assigned to each provided support service. Therefore, it is important to note that the present evaluation does not represent the number of actually provided support services during this period.

In terms of data quality, it must be noted that, in the course of the evaluation, several problems regarding the data infrastructure and data quality became apparent. Since this was a retrospective data analysis, it was not possible to influence the quality of the raw data. There is a possibility that different data entry practices between social workers may be reflected in the results. Incompleteness and incorrect entries cannot be ruled out. Similarly, only services for which a justification was given (excluding “no information required”) were included in this study. A further conspicuous feature emerged when looking at the justifications given by gender. Data entries in the male category with the justification “pregnancy of the minor” were found. This obvious misentry was probably due to an input error or unclear coding schemes, and it would be conceivable that this entry was intended to note the paternity of a minor. Furthermore, when analyzing the dataset, a considerable number of cases where no justification was given was found. These cases made up about 40% of all cases. These cases were not included in the further analysis.

Moreover, to circumvent the problem of the presumed strong dependence on the support service justifications, justification classes (caregiver vs. child level) were formed. However, the classification was largely based on the semantic content of the justifications and should be validated in future work. Potential ambiguities, especially related to parental overload, are already described above.

Since services located in the “not specified” age category and the justifications assigned to them represent an outlier, the need for a way to check whether these were people who were already of age and whose previous services were not closed in the case documentation system emerges. If this was the case, it might be useful to exclude this age category in future evaluations to counteract any subsequent bias in the results.

Furthermore, a side objective of the present work was the investigation of the feasibility of analyzing administrative data from the Child and Youth Welfare Service. Therefore, only data from 2021 were used for the time being. However, in order to gain a more comprehensive understanding of trends in justification reporting, future work should also analyze data further back in time. Therefore, the data retrieval should be automated, while the analysis steps can be performed based on the present script.

The results of the present work indicate that data quality depends on the data entry practices of the social workers. To improve the quality of the data, it seems necessary to evaluate the data entry practices between the district administrative authorities and to subsequently develop quality assurance measures. In this regard, standardized data entry represents the basis for evidence-based research, and in this regard, several studies [54,55,56,57] can be referred to that address the topic in depth. A first step would be to either develop new instruments for data collection or to use existing, well-validated instruments and validate them for use in Austria.

Not only the form of child welfare endangerment but also its intensity and duration as well as numerous other personal and socioeconomic factors influence, on the one hand, the decision to provide support and thus the further support process and, on the other hand, the further individual personal development of the persons concerned, including their mental and physical health, their quality of life, and their prospects. However, the available data are limited exclusively to the form of child welfare endangerment in the sense of a justification to be given when a service is provided at a district administrative authority, which is why no further differentiated conclusions can be drawn. Even though no socioeconomic data were available, limited generalizability is not to be assumed since it was not a sample of the data but the entirety of the data that were evaluated. For further interpretation of the results, access to socioeconomic variables would be beneficial in order to evaluate which groups are particularly at risk in the future. It is known from previous studies that families who live in poverty are more likely to be involved with child and youth welfare [58]. A review also shows that (especially black) race and low socioeconomic status are factors that contribute to overrepresentation in child and youth services [59]. Therefore, personal and socioeconomic factors should be included in a future case-based survey.

Further, it can be assumed that the children and adolescents concerned are or were often not exclusively exposed to an isolated form of child welfare risk. Thus, it seems reasonable to collect combined aspects of risk in the sense of a profile as an indicator of developmental prognosis [60]. In order to consider relevant factors as well as developments in the course of support beyond the currently available data, the planned future use of a standardized questionnaire or assessment form as a survey instrument within the framework of social diagnostics by social work professionals will also be of central importance in the field of research as well as for the evaluation of child and youth welfare measures. Additionally, gender sensitivity, especially regarding assessments, should be integrated more into social work education in the future since gender differences tend to be reflected in the types of child endangerment (i.e., justifications). Gender-specific prevention programs (e.g., violence prevention and drug prevention) should be considered at a political level.

## 5. Conclusions

With the first evaluation of the administrative data of the Lower Austrian Child and Youth Welfare Service, a foundation for more detailed analyses was created, and approaches for dealing with limitations were presented. Thus, this work provides a first basis for evidence-based and data-driven research in the field of Lower Austria’s child and youth welfare. It also became apparent that most child and youth welfare cases originate from the parents or caregivers and that there are differences in the support service justifications regarding the gender and age of the children that should be considered in future policy decisions affecting the safety and welfare of children. Future research should focus on what preventive measures can be taken on the part of caregivers in general and specifically to avoid parental overload. In addition, the research focus should be on the increasing child-level justifications as children get older in order to possibly preclude the involvement of child and youth welfare by intercepting the children’s issues before they become problematic and require professional support. Moreover, a practical requirement for the use of standardized survey instruments in order to evaluate risk factors and case-related developments in the course of support services arises for social work.

## Figures and Tables

**Figure 1 children-10-01376-f001:**
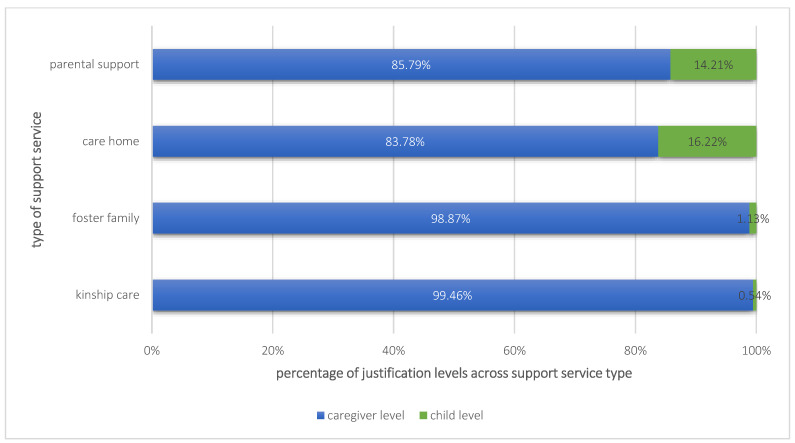
Comparison of justifications on caregiver versus child level for each support service type.

**Figure 2 children-10-01376-f002:**
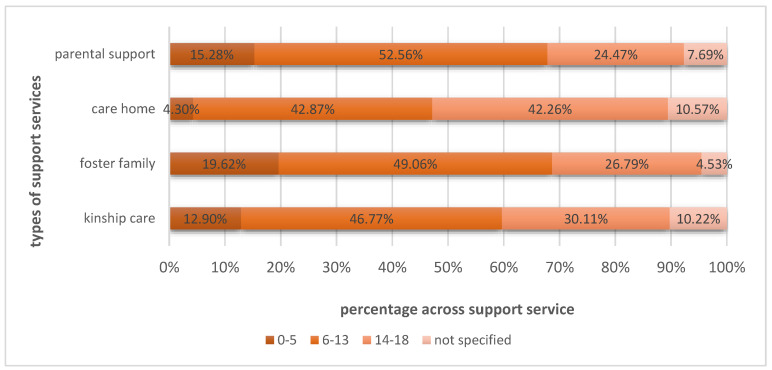
Comparison of the age group proportions across each type of support.

**Figure 3 children-10-01376-f003:**
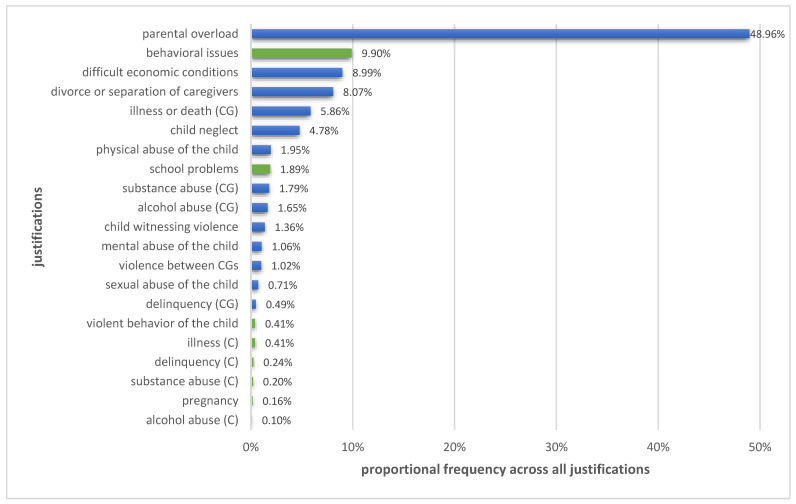
Frequencies of the unclassified justifications across all justifications for ongoing support services in 2021.

**Figure 4 children-10-01376-f004:**
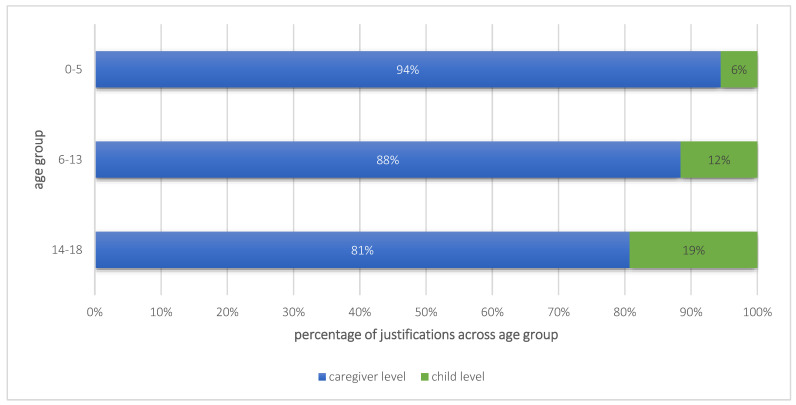
Proportional comparison of the justifications on caregiver versus child level per age group.

**Table 1 children-10-01376-t001:** Classification of the justifications into caregiver or child level.

Caregiver Level	Child Level
alcohol abuse of caregiver(s)	alcohol abuse of the child
violence between caregivers	violent behavior of the child
illness or death of caregiver(s)	illness of the child
parental overload	school problems or violation of compulsory attendance
delinquency of caregiver(s)	pregnancy of the minor
substance abuse of caregiver(s)	delinquency of the child
unfavorable economic conditions	substance abuse of the child
divorce or separation of caregiver(s)	behavioral issues
mental abuse of the child	
physical abuse of the child	
sexual abuse of the child	
child witnessing violence	
child neglect	

**Table 2 children-10-01376-t002:** Gender proportion across each support service justification.

	Male		Female		Total
Justification	*n*	%	*n*	%	*n*
parental overload	1338	56%	1070	44%	2408
behavioral issues	305	63%	182	37%	487
difficult economic conditions	230	52%	212	48%	442
divorce or separation of caregiver(s)	195	49%	202	51%	397
illness or death of caregiver(s)	134	47%	154	53%	288
child neglect	138	59%	97	41%	235
physical abuse of the child	53	55%	43	45%	96
school problems	53	57%	40	43%	93
substance abuse of caregiver(s)	53	60%	35	40%	88
alcohol abuse of caregivers(s)	39	48%	42	52%	81
child witnessing violence	35	52%	32	48%	67
mental abuse of the child	25	48%	27	52%	52
violence between caregivers	29	58%	21	42%	50
sexual abuse of the child	8	23%	27	77%	35
delinquency of caregiver(s)	12	50%	12	50%	24
illness of the child	10	50%	10	50%	20
violent behavior of the child	14	70%	6	30%	20
delinquency of the child	12	100%	0	0%	12
substance abuse of the child	6	60%	4	40%	10
pregnancy of the minor	2	25%	6	75%	8
alcohol abuse of the child	2	40%	3	60%	5
total	2693	55%	2225	45%	4918

**Table 3 children-10-01376-t003:** Proportional comparison of the individual justifications across age groups.

	Parental Overload		Behavioral Issues		Difficult Economic Conditions		Divorce or Separation of Caregiver(s)		Death or Illness of Caregiver(s)		Child Neglect		Physical Abuse of the Child		School Problems		Substance Abuse of the Caregiver(s)		Alcohol Abuse of the Caregiver(s)		Child Witnessing Violence	
	*n*	%	*n*	%	*n*	%	*n*	%	*n*	%	*n*	%	*n*	%	*n*	%	*n*	%	*n*	%	*n*	%
0–5	381	56.95%	25	3.74%	71	10.61%	31	4.63%	39	5.83%	38	5.68%	4	0.60%	4	0.60%	27	4.04%	11	1.64%	11	1.64%
6–13	1237	49.76%	215	8.65%	261	10.50%	192	7.72%	145	5.83%	124	4.99%	53	2.13%	57	2.29%	45	1.81%	36	1.45%	40	1.61%
14–18	622	45.57%	194	14.21%	85	6.23%	112	8.21%	78	5.71%	63	4.62%	30	2.20%	32	2.34%	14	1.03%	25	1.83%	15	1.10%
not specified	168	42.21%	53	13.32%	25	6.28%	62	15.58%	26	6.53%	10	2.51%	9	2.26%	0	0.00%	2	0.50%	9	2.26%	1	0.25%
total	2408	48.96%	487	9.90%	442	8.99%	397	8.07%	288	5.86%	235	4.78%	96	1.95%	93	1.89%	88	1.79%	81	1.65%	67	1.36%
	**Mental abuse of the child**		**Violence between** **caregiver(s)**		**Sexual abuse of the child**		**Delinquency of caregiver(s)**		**Illness of the child**		**Violent behavior of the child**		**Delinquency of the child**		**Substance abuse of the child**		**Pregnancy of the minor**		**Alcohol abuse of the child**		**Total**	
	** *n* **	**%**	** *n* **	**%**	** *n* **	**%**	** *n* **	**%**	** *n* **	**%**	** *n* **	**%**	** *n* **	**%**	** *n* **	**%**	** *n* **	**%**	** *n* **	**%**	** *n* **	
0–5	1	0.15%	12	1.79%	4	0.60%	2	0.30%	2	0.30%	0	0.00%	0	0.00%	0	0.00%	6	0.90%	0	0.00%	669	
6–13	17	0.68%	18	0.72%	14	0.56%	16	0.64%	7	0.28%	6	0.24%	2	0.08%	0	0.00%	0	0.00%	1	0.04%	2486	
14–18	24	1.76%	16	1.17%	15	1.10%	3	0.22%	11	0.81%	9	0.66%	7	0.51%	8	0.59%	0	0.00%	2	0.15%	1365	
not specified	10	2.51%	4	1.01%	2	0.50%	3	0.75%	0	0.00%	5	1.26%	3	0.75%	2	0.50%	2	0.50%	2	0.50%	398	
total	52	1.06%	50	1.02%	35	0.71%	24	0.49%	20	0.41%	20	0.41%	12	0.24%	10	0.20%	8	0.16%	5	0.10%	4918	

## Data Availability

Restrictions apply to the availability of these data. Data were obtained from Land Niederösterreich and are available from the corresponding author with the permission of Land Niederösterreich.

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
