# Peer review of "Why Are Child and Youth Welfare Support Services Initiated? A First-Time Analysis of Administrative Data on Child and Youth Welfare Services in Austria"

_children, 2023, doi:10.3390/children10081376_

Round 1

Reviewer 1 Report

The study analyzed reasons for child and youth welfare services initiation and distribution by age and gender. Iyt presents valuable data. In general, the study could have used more rigorous statistical techniques. The type of statistics provided are generally covered by the administrative reports created and published from such administrative datasets.

The author did not even have the raw dataset. Instead, they used the aggregated data to report aggregated data. This does not meet the rigor and standards for publication in the MDPI’s Children journal.

 The authors should comment in the last paragraph of the Introduction section or the first paragraph of the Methods section should justify why the analysis was limited to an “exploratory” level and why the variation and disparities in the utilization of the child and youth welfare support services are not examined. If the ongoing administrative data are collected, are there any studies from this data, if not, for what reasons/challenges? Exploratory research can be justified when a problem is not clearly defined, has been under-investigated, or is otherwise poorly understood. Exploratory research is justified in examining an issue more thoroughly before attempting to quantify mass responses into statistically inferable data. It can also be used to investigate a problem that is not clearly defined or test the feasibility of starting a more in-depth study. None of that is hardly applicable to the current situation, but if it is, the authors should state that and build their case.

Specific comments

Title: The authors should consider rewording the title as “and their justifications” seems out of place.

Abstract:

 The abstract is missing some details. For instance, some details about the study design and dataset are essential for readers to understand the scope and scale of the study, including the dataset's complete name, the agency that collects information, and the secondary/primary use of the data. In addition, the abstract lacks detail about the analysis techniques.

Introduction: The introduction section is seriously flawed. The authors present very little, if any, literature on the topic—reasons and factors associated with the initiation (or lack thereof of) child and youth welfare support services. On the other hand, the authors dedicate roughly half of the Introduction section (i.e. 318 words out of the total 662-word introduction section) to justify the use of the secondary/administrative data.

 The authors need to dedicate more effort to documenting the existing literature on the topic and identifying the gaps in the research, to justify the current study, particularly the exploratory nature.

 Data, Method, and Statistical Analysis

The authors mentioned the source of data which is Child and Youth Welfare (SZF). However, the raw data were not obtained that would have facilitated a more nuanced analysis.  An ambiguity about data is that the authors had access to summary data but they mentioned that they added reasons amnaually. How did they link the aggregared data with the individual level reasons is not clear. The authors limited the data to child welfare endangerment and ignored the contextual factors (personal and socio-economic) that might affect the support process. In limitation, authors should add details about the potential influence of contextual factors on the generalizability and interpretation of the findings.

Insufficient information about the data analysis: Authors must write about the statistical tests or methods used to analyze the data and derive the key findings. For example, they mentioned that they “clustered the reasons” but do not describe the methods of clustering (e.g., were factor analysis or a modification of that procedure used?).

 The authors have presented a comparison of the proportions of the individual justifications across age groups. However, the comparison would have been only meaningful if some statistical test of significance was performed. Therefore the discussion of such differences is not much meaningful in terms of their implications for the program or its stakeholders. In addition, the comparison by race and other demographic characteristics should have been included.

 The description of the analytical methods is also vague. For instance, the authors write: “We carried out 481 separate queries individually, based on …type of support service to obtain all relevant service combinations.” A description of data collection, data enhancement, or statistical methods is not clear to the reader.

Results

Authors should provide more details about the possible causes behind these differences, and their implications would add depth to the interpretation of the results.

 No information about the social and cultural factors that might influence the observed pattern has been presented. Authors should provide contextual information within the broader social context of child and youth welfare in Lower Austria.

Author Response

Dear reviewer, 

thank you for all the time and effort you put into thoroughly reviewing our paper! Please find attached the response letter to your review. 

Best regards, 
the authors

Reviewer 2 Report

The authors highlight the importance of examining administrative data to inform trends in the child and youth population involved the welfare system. While acknowledging the limitations of using administrative data for research purposes, the authors show that much can be learned from examining such data. In this case, the data revealed that parental/caregiver factors were a major contributor to children and youth's involvement in the welfare system, with "parental overwhelm" playing a significant role.  

The problem statement, materials and methods, results, and discussion sections are adequately described.  

The level of analysis of the data is at the descriptive level which revealed the differences and trends presented in the paper.  Did the authors attempt any exploratory  inferential statistical analysis to identify any statistical significance in the differences and trends noted in the data?    

In the discussion section, the authors review the various trends they observed in the data, through the cross tabulations of gender and support service justification; age and contributing factor for welfare involvement, for example. They provide literature support for these observations by citing other studies and reports of statistics (from Austria).  Given these findings,  the role, the methods, and decision making of the social work professionals who make the decisions in regard determining the causes as well as the solutions to the problem becomes very important to examine.  For example,  the authors indicate the need for use of standardized questionnaire as way of attempting to insure that there is some standardization in the assessment.  Are there other recommendations that the authors can make to strengthen social work professionals' skills in assessment, particularly as they related to taking into account gender and age in determining support services? 

How do the results reflect Lower Austria's social welfare's  vision and mission for its children and youth?  From the data, it appears that relatively small percentage of children and youth end up in care that are not with their family of origin. If true, is this finding a "positive" reflection of the agency's overall mission of providing care in the least restrictive setting for the child or youth- with his or her family-preserving family connections and relationships?  

minor editing around word choice and grammar would make the manuscript clearer.  Phrases such as " Parental Overwhelm" is an example of a term used  that would benefit from being reworded as it is not a common term used in English. 

Author Response

Dear reviewer, 

thank you for taking the time to review our paper. We appreciate the time and effort you put into it. Please find attached the response letter to your review. 

Best regards,
the authors

Reviewer 3 Report

The paper explores the reasons why child and youth welfare was initiated and whether the justification was more at the level of the caregiver or the children or adolescents themselves in Lower Austria. The authors use administrative sources which were prepared for data analysis aiming to respond to the posited research question. One of the main contributions of the paper is that it outlines the strengths and the challenges in using administrative data in this field of research and makes recommendations how to improve the completeness and quality of the administrative data sources. The paper demonstrates that administrative data are an important source of information and the introduction of quality standards have string potential for studies of child and youth welfare. My suggestions to the authors are the following:

1. It should be explained in brief what are the existing sources of data on child and youth welfare in Austria. For example, are there any official reports (administrative statistics) published regularly that aggregate the records of the social workers?

2. Are there any recommendation for inclusion of additional information in the records of the social workers apart from that already available characteristics of children and caregivers?

3. The comparison of the results with other sources of information (surveys like ESPAD and Austrian prevalence study on violence against women and men, etc. is very interesting. The authors should elaborate in a little bit more details how the administrative data sources and surveys can complement each other, e.g. both can be used for cross-validation purposes, for studied of specific groups of particular interest, etc.

4. It is not clear how the data quality was evaluated in the process of selection and re-thriving of information from the administrative records . Are there any specific criteria and cleaning rules that were applied in the selection of data for the analysis of child and youth welfare?

Author Response

Dear reviewer, 

thank you very much for taking the time to review our paper and providing valuable comments on it. Please find attached the response letter to your review. 

Best regards,
the authors

Reviewer 4 Report

Thank you for this opportunity to revise the manuscript titled "Why are child and youth welfare support services initiated? A first-time analysis of administrative data on child and youth welfare services in Austria and their justifications" that was submitted to Children. The relevance of the research realized by the authors of the paper is obvious.

I leave several comments about the manuscript which are listed below:

1. The introduction should state the contribution to the literature.

2.  The paper could mention the indirect costs of mental health problems 

3. The Conclusion section could discuss more about the practical lessons that stem from the estimation results that are presented in the paper.

These thought is not of a fundamental nature, but requires author response.

This paper holds actual value to the readers on Children (Special Issue

Research on Child Trauma and Protection).

I will be glad to review the revised manuscript.

Author Response

Dear reviewer, 

thank you very much for taking the time to review our paper and for your valuable comments on it. Please find attached the response letter to your review.

Best regards,
the authors

Round 2

Reviewer 1 Report

Most of my comments about the unit of analysis and lack of rigor in methodology were justified based on the authors' lack of access to raw data. I have suggested no additional changes because the limitations I indicated in Round 1 seem beyond the authors' control. 

Author Response

We hope to have implemented your valued feedback to your satisfaction. Thank you for your review!
